# Microvascular Imaging of Hepatic Hemangiomas

**DOI:** 10.3390/diagnostics15222917

**Published:** 2025-11-18

**Authors:** Hakan Baş, Süleyman Filiz

**Affiliations:** 1Department of Radiology, Gazi Mustafa Kemal Occupational and Environmental Diseases Hospital, Silahtar Street No:6, Emniyet, Yenimahalle, 06560 Ankara, Türkiye; suleymanfilizz@gmail.com; 2Department of Radiology, Faculty of Medicine, Ufuk University, Mevlana Boulevard No:86-88, Balgat Campus, 06520 Ankara, Türkiye

**Keywords:** ultrasonography, hemangioma, liver neoplasms, microvascular imaging, ultrasound imaging

## Abstract

**Background/Objectives:** We aimed to characterize the microvascular imaging (MI) to demonstrate in hepatic hemangiomas in routine practice and to quantify the impact of lesion depth on MI signal detectability, and—when present—describe the distribution of MI appearances. **Methods:** In this single-center, retrospective study from January 2021 to December 2023, we screened 91 patients with 121 focal hepatic lesions on ultrasound. Lesions without typical hemangioma enhancement on dynamic MRI or dynamic CT were excluded. Two radiologists independently assessed MI signals and patterns using the Jeon classification, blinded to clinical and CT/MRI data; inter-observer agreement was quantified with Cohen’s κ. **Results:** Of 121 screened lesions, 36 lacked typical enhancement and were excluded; 85 hemangiomas remained. A total of 13 were excluded for motion artifacts near the heart or pulsatile vessels, yielding 72 hemangiomas (61 patients) for analysis. No lesion showed flow on color or power Doppler. MI signals were detected in 68/72 hemangiomas (94.4%). Among signal-positive lesions (*n* = 68), the patterns were non-specific in 25.0% (17/68), nodular rim in 22.1% (15/68), strip rim in 17.6% (12/68), central dot-like in 16.2% (11/68), peripheral dot-like in 10.3% (7/68), and staining in 8.8% (6/68). Signal-negative lesions were deeper than signal-positive lesions (median depth: 85 mm vs. 41.5 mm; *p* < 0.05). The inter-observer agreement was very good (κ = 0.821, 95% CI 0.767–0.921). **Conclusions:** MI is a reproducible, contrast-free technique that demonstrates hemangioma vascularity with high detection rates, particularly in more superficial lesions. In this cohort, lesion depth rather than size was the primary determinant of MI signal detectability. MI should be considered complementary to CT/MRI and may be especially useful where contrast agents are unavailable or contraindicated.

## 1. Introduction

Ultrasound (US) is widely used as the first-line imaging modality for focal hepatic lesions (FHLs) due to its accessibility, cost-effectiveness, and noninvasive nature [1,2,3,4,5]. Among FHLs, hepatic hemangiomas are the most common benign tumors, often detected incidentally [2,4]. Classical imaging modalities such as contrast-enhanced computed tomography (CECT) and magnetic resonance imaging (MRI) remain the gold standards for accurately diagnosing hemangiomas through their characteristic centripetal enhancement pattern [6,7]. However, some patients may not be suitable candidates for repeated contrast-enhanced imaging due to contraindications or safety concerns.

Focal liver lesions frequently require further imaging or multidisciplinary evaluation to achieve definitive diagnosis [8]. Accurate characterization of hepatic hemangiomas remains limited by modality-specific constraints that impact diagnostic confidence and clinical workflow. Dynamic contrast-enhanced CT demonstrates variable sensitivity, particularly in subcentimeter lesions, and is frequently compromised by suboptimal arterial phase timing and inter-observer variability. The use of contrast agents is associated with risks such as nephrotoxicity and hypersensitivity reactions [9,10]. Repeated CT examinations contribute to cumulative radiation exposure and raise safety concerns, particularly in younger patients and those under long-term surveillance [11]. Although MRI provides high soft-tissue contrast and diagnostic performance, its utility is curtailed by patient-related contraindications, prolonged acquisition times, and elevated cost [12,13]. B-mode ultrasound, despite its accessibility, exhibits reduced sensitivity in steatotic livers and deep lesions, and its diagnostic yield remains highly operator-dependent; furthermore, conventional Doppler fails to detect low-velocity microvascular flow typical of hemangiomas [5,14,15,16]. CEUS offers excellent diagnostic accuracy but necessitates intravenous contrast, is susceptible to motion artifacts, and requires dedicated expertise and infrastructure, all of which limit its generalizability [2,4,8,17,18]. Atypical enhancement patterns may mimic hypervascular malignancies, increasing the likelihood of unnecessary follow-up or intervention [19]. Despite structured reporting initiatives such as CEUS LI-RADS, variability in diagnostic thresholds and institutional practice patterns persists [8]. Geographic and economic disparities limit access to advanced imaging technologies in many clinical settings, especially in underserved or rural populations. These limitations underscore the need for accurate, contrast-free, and widely deployable alternatives in hepatic lesion assessment. Microvascular imaging (MI) is an advanced Doppler technique that uses adaptive clutter suppression and high-frame-rate processing to depict slow intralesional and perilesional flows without contrast. By minimizing wall-filter losses and respiratory/pulsatile motion contamination, MI increases the conspicuity of low-velocity signals typically missed by conventional color or power Doppler, enabling real-time vascular phenotyping in the liver. From a clinical standpoint, MI signal visualization (and descriptive appearances when present) is intended to complement, rather than replace, the reference-standard diagnosis on CT/MRI. Rim-type and dot-like appearances and homogeneous “staining” on MI should be interpreted together with B-mode features and clinical context to avoid misclassification with hypervascular mimics (e.g., focal nodular hyperplasia). In settings where contrast agents are unavailable or contraindicated, such structured, contrast-free pattern information can support triage and follow-up decisions [20,21,22,23].

Therefore, we conducted a single-center retrospective study to characterize microvascular imaging (MI) findings in hepatic hemangiomas using a standardized acquisition protocol. The study aimed to describe the distribution of vascular patterns and to assess how technical and lesion-related factors, particularly the skin-to-lesion depth, affect MI signal detectability while accounting for lesion size and segmental location. The inter-observer agreement for MI-based classification was also evaluated to determine reproducibility. We hypothesized that greater lesion depth, rather than lesion size, would be associated with lower odds of detecting MI signals because of ultrasound beam attenuation and motion artifacts in deeper regions of the liver [24,25].

## 2. Materials and Methods

### 2.1. Study Design and Ethics

We conducted a single-center retrospective study of patients evaluated between January 2021 and December 2023. The institutional review board approved the study (Approval No. 19; 12 February 2024), and informed consent was waived owing to the retrospective design and use of de-identified data.

### 2.2. Patient Selection

We screened 91 patients with 121 focal hepatic lesions identified on ultrasonography (US). The reference-standard dynamic imaging modality was contrast-enhanced MRI in 50 patients and multiphasic CT in 41. MRI was prioritized when iodinated-contrast allergy or broader safety considerations such as renal risk were present, whereas CT was selected for MRI-incompatible implants, claustrophobia, or when urgent scanner availability was required. Hepatic hemangioma (HH) was defined by typical dynamic enhancement on MRI or CT, characterized by peripheral, nodular, discontinuous arterial enhancement with progressive centripetal fill-in on the portal venous and delayed phases. Following expert review of the dynamic studies, 36 lesions that did not demonstrate this characteristic pattern were excluded. The eligible cohort comprised 61 patients with 85 hepatic hemangiomas. Thirteen hemangiomas situated adjacent to the heart or large pulsatile vessels were further excluded because motion artifacts precluded reliable MI assessment. The final analytic sample therefore consisted of 72 hemangiomas. Multiple lesions per patient were permitted, and each target lesion served as an independent unit of analysis.

### 2.3. Image Acquisition

All ultrasound examinations were performed using the RS85 Prestige system (Samsung Medison, Seoul, Republic of Korea) equipped with a convex transducer (CA1-7A, 1–7 MHz). A board-certified radiologist with over ten years of experience in hepatic imaging conducted all acquisitions. Patients were examined in the supine or left lateral decubitus position, and images were obtained during suspended respiration to minimize motion artifacts. For each lesion, the segmental location, maximal diameter, echogenicity (hypoechoic, isoechoic, or hyperechoic), and distance from the skin surface to the lesion center were recorded. Color and power Doppler examinations were first obtained using a standardized hepatic preset, followed by MI. MFlow, the vendor-specific implementation of MI, was applied under identical technical conditions for all cases. Within the general abdominal preset, the velocity range for flow detection was automatically calibrated between −2.3 cm/s and +2.3 cm/s. Flow intensity was visualized using a monochromatic power-mode scale provided by the application. A manually defined Doppler region of interest was adjusted to encompass the entire lesion, and imaging was performed in a split-screen format that simultaneously displayed B-mode and MI views. To achieve optimal visualization of flow within the target area, acquisition settings were standardized across all examinations: frame averaging of 5–7, tissue suppression level 2–3, filter level 2–3, sensitivity 26–32, dynamic range 25–35, balance 20–30, and output power 80–90%. Each lesion was recorded as a 3–5 s cine loop obtained during suspended respiration and exported in DICOM format for subsequent blinded evaluation and analysis. Accumulation imaging modes were not used, and all other machine parameters were maintained identical to ensure reproducibility and inter-case comparability.

### 2.4. Image Analysis

All cine loops were de-identified and independently reviewed by two radiologists with ten and seven years of experience in abdominal imaging. Both reviewers were blinded to the patients’ clinical information and to the results of computed tomography and magnetic resonance imaging. The primary endpoint of the analysis was the presence or absence of microvascular imaging signals. These signals were defined as Doppler-derived slow-flow vascular signals originating from intra- or perilesional microvessels, visualized by the motion-suppression algorithms of the microvascular imaging technique. This definition was purely imaging-based and did not imply histopathological confirmation. When a microvascular imaging signal was identified, the vascular pattern was classified according to a previously published schema without any modification. The patterns were described as follows: strip rim pattern, characterized by a few curved peripheral vessels; nodular rim pattern, defined by peripheral curved vessels with scattered dot-like bulges; peripheral dot-like pattern, showing multiple distended peripheral dots; central dot-like pattern, consisting of distended dots predominantly located at the center of the lesion; non-specific pattern, showing small and irregular vessel clusters without a particular arrangement; and staining pattern, defined by homogeneous intralesional signal distribution. All disagreements between the two observers were reviewed jointly, and the final pattern classification was determined by consensus. Figure 1 illustrates these patterns.

### 2.5. Statistical Analysis

Analyses were performed at the lesion level. Continuous variables are summarized as medians with interquartile ranges or as means with standard deviations according to distribution, which was assessed by visual inspection of histograms and by the Shapiro–Wilk test. Group comparisons for continuous variables used the Mann–Whitney U test when distributions were non-normal and the independent-samples *t* test when normality was satisfied. Categorical variables are summarized as counts and percentages and were compared using the chi-square test or Fisher’s exact test as appropriate. The inter-observer agreement for the presence of a microvascular imaging signal and for pattern categorization was quantified with Cohen’s kappa together with ninety-five percent confidence intervals. All tests were two-sided with a significance level of 0.05. All analyses were conducted with SPSS version 22.

## 3. Results

The study population comprised 61 patients, and a total of 85 hemangiomas were initially detected. A total of 13 hemangiomas in segments close to the heart and pulsatile vessels were excluded due to motion artifacts, leaving 72 hemangiomas for final analysis. The flowchart of the study is demonstrated in Figure 2.

Among the 72 included hemangiomas, five measured ≤1 cm in maximum diameter (8 mm in one lesion and 10 mm in four lesions). The median (IQR) size of all the hemangiomas was 24.5 (16–37) mm. The median (IQR) length of the skin-to-lesions was 43 (28–62) mm. The lesions exhibited a distribution of echogenicity characterized as hypoechoic (*n*: 6, 8.3%) and hyperechoic (*n*: 66, 91.7%).

No apparent signal was obtained from color Doppler imaging (CDI) and power Doppler imaging (PDI) in any lesion. In 4 (5.6%) hemangiomas, the MI signal was undetectable. Among MI-positive hemangiomas (*n* = 68), the pattern frequencies were non-specific in 25.0% (17/68), nodular rim in 22.1% (15/68), strip rim in 17.6% (12/68), central dot-like in 16.2% (11/68), peripheral dot-like in 10.3% (7/68), and staining in 8.8% (6/68). The predominant pattern was the non-specific vascular pattern, succeeded in descending order by the nodular rim pattern and the strip rim pattern. The median (IQR) skin-to-lesion distance in the 4 hemangiomas without signal (85 [65.5–102] mm) is statistically significantly greater than that of the other 68 hemangiomas (41.5 [27.5–59] mm) (*p* < 0.05). Nonetheless, there was no statistically significant variation in lesion dimensions between the cohort demonstrating a signal and the cohort not presenting a signal. The microvascular imaging characteristics are summarized in Table 1. The inter-observer concordance for the categorization of MFlow characteristics between the two operators was very good (κ = 0.821 95% CI: 0.767–0.921).

## 4. Discussion

In this single-center retrospective cohort, microvascular imaging (MI) detected slow-flow vascular signals in most hepatic hemangiomas, whereas color and power Doppler showed no flow. Among signal-positive lesions, the non-specific pattern was most frequent, followed by nodular rim and strip rim appearances. Lesions without an MI signal were substantially deeper than those with a signal, while the lesion diameter was comparable between groups. Under uniform acquisition presets and without the use of accumulation imaging, these data indicate that skin-to-lesion depth is the principal factor limiting MI signal capture in the liver, most plausibly due to frequency-dependent attenuation and motion artifacts at greater depths [26,27].

Prior pattern-based MI studies, including Jeon and colleagues, most often described the strip rim configuration and reported lower overall detection rates [4]. In our cohort, the non-specific configuration was more frequent, and the overall detection rate was 94.4%. Several factors may account for these differences. We used uniform presets with a narrow velocity scale and fixed suppression and filter ranges. Accumulation modes were not applied, which can change the conspicuity of thin peripheral tracks. Our case mix also included several deep-seated lesions, and in our data, lesion depth showed the strongest association with non-detection [28,29,30]. Vendor-specific implementations of clutter suppression may further contribute to inter-study variability. Taken together, these technical and sampling considerations may explain the shift in pattern hierarchy and support a cautious view of MI as a contrast-free adjunct for depicting hemangioma vascularity in routine practice [20,26].

Direct histopathology is seldom available for typical hepatic hemangiomas, yet several imaging–pathophysiology links are plausible and help anchor the observed MI appearances. Strip rim and nodular rim configurations may reflect slow venous flow within peripheral vascular lakes and marginal channels, which broadly parallels the early peripheral nodular enhancement seen on dynamic CT or MRI. Peripheral dot-like foci may correspond to tangentially imaged microchannels at the lesion–parenchyma interface, where the point-spread function accentuates punctate signals. Central dot-like foci could arise from intralesional vascular lakes or trabecular crossings that contain sluggish flow detectable after clutter suppression. The staining configuration on MI denotes a relatively homogeneous intralesional slow-flow signal and should not be equated with the contrast-enhanced staining phenomenon described in focal nodular hyperplasia, as the underlying physics and diagnostic implications differ. In this manuscript, “MI signal” refers to a clutter-suppressed Doppler depiction of very-low-velocity flow originating from intra- or perilesional microvessels and is presented as an imaging construct rather than histologic proof of vascular architecture [4,29,31,32,33,34,35,36,37,38].

Microvascular imaging (MI) provides a contrast-free, real-time depiction of very-low-velocity blood flow, thereby clearly demonstrating vascularity in hepatic hemangiomas. Relative to biopsy, MI enables whole-lesion vascular assessment without procedural risk; compared with MRI, it offers immediate bedside evaluation and repeatability without device or renal limitations, although MRI remains superior for depth-independent, multiparametric tissue characterization. The technique markedly outperforms conventional Color or Power Doppler, which usually fails to display intralesional flow despite the histologically confirmed hypervascularity of hemangiomas [39,40]. In the histology-proven series of Dietrich et al., feeding or draining vessels were seen in 25 of 58 lesions (43%), homogeneous hypervascularity in 4 (7%), and no Doppler signal in 29 (50%) [41]. In our cohort, CDI and PDI did not show intralesional flow, whereas MI demonstrated a detectable signal in 94.4% of hemangiomas. These findings reflect descriptive differences observed in this dataset and should not be interpreted as a formal assessment of superiority, as the study was not designed for performance comparison. Compared with Contrast-Enhanced Ultrasound (CEUS), MI avoids microbubble administration and intravenous access yet complements CEUS’s well-validated enhancement criteria—peripheral nodular arterial enhancement (74–91%) and centripetal fill-in within about 180 s [41,42]. Overall, MI bridges the diagnostic gap between flow-negative Doppler and contrast-requiring CEUS, offering an accessible, repeatable, and cost-effective adjunct for screening and follow-up, especially when contrast use is contraindicated or unavailable.

Clinical implications follow from these observations but warrant cautious framing. Multiphasic contrast-enhanced CT or MRI remains the accepted reference standard for typical hemangioma and enables confident noninvasive diagnosis [43,44]. Where available, contrast-enhanced ultrasound provides high diagnostic accuracy and adds dynamic perfusion information that is not captured by non-contrast techniques [45]. In routine practice, microvascular imaging should be used as a complement: it may increase confidence when B-mode features are typical yet conventional color or power Doppler is negative, and it is particularly relevant when contrast administration is unavailable or contraindicated due to factors such as severe contrast allergy, limited access or cost, or MRI-incompatible devices. The finding in our cohort that CDI/PDI were uniformly negative, while MI frequently demonstrated slow flow, is consistent with the known limitations of conventional Doppler for very-low-velocity microvascular signals and with the motion-suppression and high-frame-rate strategies that underlie MI [46,47,48,49,50,51].

This study has several methodological strengths. It was conducted in a randomly selected patient cohort that represented routine clinical practice, without any restriction on lesion size or skin-to-lesion depth. Lesions showing typical peripheral nodular enhancement with progressive centripetal fill-in on reference-standard CT or MRI were included, whereas atypical enhancement patterns that cannot be confidently distinguished from hypervascular metastases without histopathological confirmation, such as flash-filling hemangiomas, were excluded [52]. This approach ensured a homogeneous dataset and strengthened the methodological reliability of the results. All MI images were interpreted independently by two radiologists, and the very good inter-observer agreement obtained supports the reproducibility of MI and its potential use as a complementary technique to B-mode ultrasound for the triage and follow-up of typical hepatic hemangiomas. In regions where contrast-enhanced ultrasound agents are not available, as in our country, MI may offer a time- and cost-effective alternative that may facilitate early triage and reduce reliance on CT or MRI, which are often limited by contraindications, higher cost, and longer acquisition times.

Microvascular imaging depicts very-low-velocity intra- and perilesional flow beyond the sensitivity of conventional color Doppler, thereby assisting benign–malignant discrimination [5,48]. In hepatic hemangiomas, peripheral strip rim and nodular rim configurations are frequently encountered and, when concordant with B-mode features, support a benign diagnosis; however, similar rim-type appearances may occur in metastases, limiting specificity if interpreted in isolation [48,53]. Focal nodular hyperplasia typically demonstrates a spoke-wheel arterial inflow pattern on MI/CEUS, which serves as a reasonably specific rule-in sign [30,54]. This characteristic vascular pattern has been identified as a specific diagnostic feature that helps differentiate focal nodular hyperplasia from other focal liver lesions [30,55]. Hepatocellular carcinoma (HCC) and hepatic adenomas more often exhibit chaotic or honeycomb intratumoral microvascular architecture with higher vascular indices, yet substantial pattern overlap across entities constrains stand-alone diagnostic accuracy [5,21,54]. Studies have demonstrated that HCC frequently shows more hypervascular supply patterns on superb microvascular imaging compared to benign lesions, though morphometric analysis of tumor microvessels remains an evolving area of investigation [5,21].

This study has several limitations that should be considered when interpreting the findings. Its retrospective, single-center design and lesion-level analysis limit generalizability and may reduce statistical precision despite accounting for multiple lesions per patient. The reference diagnosis relied on typical multiphasic enhancement on CT or MRI without histopathology, which is appropriate for benign hemangioma in clinical practice but precludes direct microanatomical correlation. Contrast-enhanced ultrasound was not performed because ultrasound contrast agents were not available during the study period, so same-cohort comparisons with CEUS could not be made. Lesions located near the heart or adjacent to large pulsatile vessels were excluded due to motion artifacts, which represents a practical boundary condition for MI in routine imaging. We did not quantify hepatic parenchymal factors such as steatosis or fibrosis that may influence acoustic attenuation and MI signal detectability, and we did not test accumulation imaging modes, which could alter conspicuity under selected conditions. Finally, results may vary across ultrasound platforms because vendor-specific clutter-suppression strategies differ, underscoring the need for cross-vendor validation.

Prospective, multicenter studies are needed to confirm these findings across different scanners and practice environments and to define depth thresholds that ensure reliable MI acquisition. Comparative designs that benchmark MI against CEUS and MRI should be prioritized to clarify incremental value and to characterize performance in atypical presentations, including flash-filling hemangiomas that overlap with hypervascular metastases on cross-sectional imaging. Incorporating parenchymal metrics such as steatosis or fibrosis, as well as patient-level outcomes, resource use, and time-to-diagnosis, would facilitate the assessment of clinical utility and cost-effectiveness in settings where contrast administration is limited by availability, cost, or contraindications.

In conclusion, under standardized presets, MI depicted slow-flow microvasculature in most hepatic hemangiomas, while color and power Doppler were negative. In this cohort, lesion depth was the main factor associated with MI signal non-detection, whereas lesion size showed no measurable association. The technique should be considered a contrast-free complement to B-mode ultrasound that can support the triage and follow-up of typical hemangiomas, particularly where CEUS is unavailable and cross-sectional imaging is constrained by cost, access, or clinical contraindications. Further prospective and comparative work is needed to define MI’s precise role alongside established contrast-enhanced modalities.

## Figures and Tables

**Figure 1 diagnostics-15-02917-f001:**
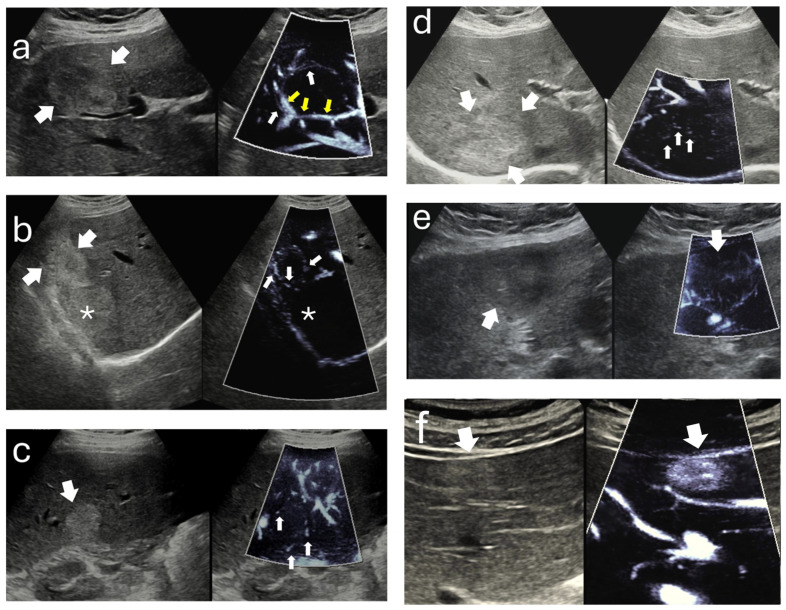
Representative MI appearances/signal distributions of hepatic hemangiomas on paired B-mode (**left**) and MFlow MI (**right**) images. (**a**) Strip rim pattern. A well-defined hyperechoic hemangioma is seen on B-mode (thick white arrows). On MI, a thin curvilinear slow-flow tract outlines the lesion margin, forming an incomplete rim (white arrows), consistent with the strip rim configuration. The yellow arrows highlight a brighter, linearly encoded vessel posterior to the lesion—an adjacent vein that courses next to, but remains separate from, the hemangioma on cine review. (**b**) Nodular rim pattern. On B-mode, two adjacent hemangiomas are visualized (thick white arrows). The more superficial lesion demonstrates the nodular rim configuration on MI, characterized by curved peripheral vessels with scattered, dot-like bulges (white arrows). The deeper lesion, marked by an asterisk, lies at greater depth from the skin surface and shows no detectable MI signal. (**c**) Peripheral dot-like pattern. B-mode demonstrates a well-defined hemangioma (thick white arrows). On MI, multiple punctate, distended foci are seen along the lesion margin (white arrows) without forming a continuous rim, consistent with the peripheral dot-like configuration. (**d**) Central dot-like pattern. B-mode shows a well-circumscribed hemangioma (thick white arrows). On MI, several clustered punctate foci are identified within the central portion of the lesion (white arrows), while the peripheral region shows comparatively less signal intensity, consistent with the central dot-like configuration. (**e**) Non-specific vascular pattern. B-mode depicts a mildly hyperechoic hemangioma (thick white arrows). On MI, small, irregular clusters of slow-flow signal are seen without a dominant rim- or center-predominant arrangement (white arrows), representing a non-specific vascular configuration characterized by heterogeneous, low-velocity microvascular flow. (**f**) Staining pattern. B-mode shows a typical hemangioma with well-defined margins (white arrows). On MI, a relatively homogeneous intralesional slow-flow signal (white arrow) fills most of the lesion without discernible vascular structures, consistent with the staining pattern.

**Figure 2 diagnostics-15-02917-f002:**
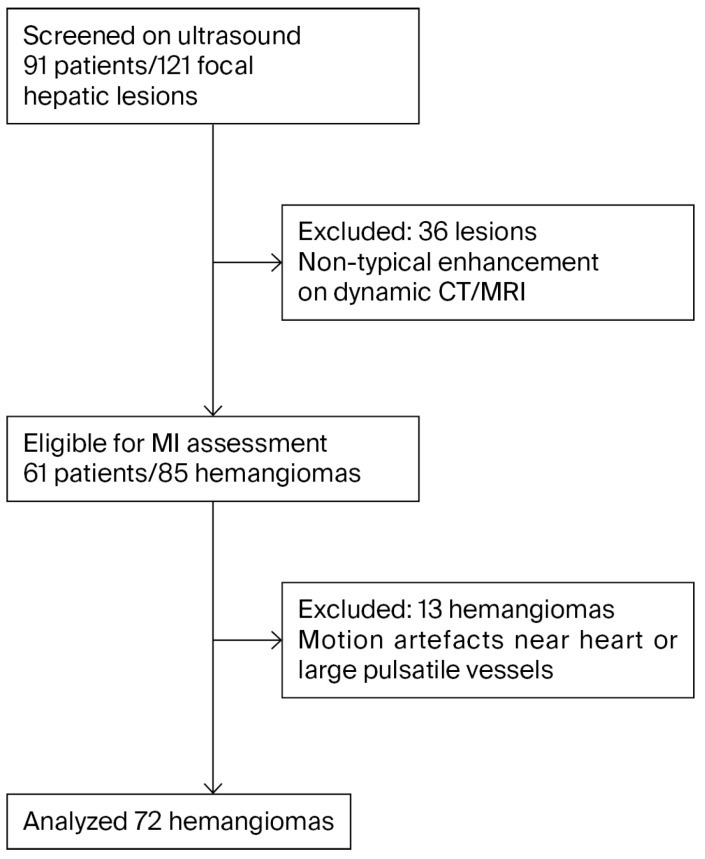
Flowchart of the study.

**Table 1 diagnostics-15-02917-t001:** Baseline lesion characteristics and microvascular imaging (MI) outcomes.

Measure	Value
**Panel A. Baseline lesion characteristics (overall, *n* = 72)**	
Lesion size, mm—median (IQR)	24.5 (16–37)
Skin-to-lesion length, mm—median (IQR)	43 (28–62)
Hypoechoic—*n* (%)	6 (8.3)
Hyperechoic—*n* (%)	66 (91.7)
**Panel B. Doppler/MI outcomes and MI signal distribution (appearance categories)**	
CDI/PDI flow present—*n* (%)	0 (0.0)
MI signal present—*n* (%)	68 (94.4)
MI signal absent—*n* (%)	4 (5.6)
Depth by MI signal (mm), median (IQR)	No MI signal: 85 (65.5–102) vs. MI-positive: 41.5 (27.5–59); *p* < 0.05
**MI patterns (calculated within MI-positive lesions, *n* = 68)**	
Non-specific—*n* (%)	17 (25.0)
Nodular-rim—*n* (%)	15 (22.1)
Strip-rim—*n* (%)	12 (17.6)
Central dot-like—*n* (%)	11 (16.2)
Peripheral dot-like—*n* (%)	7 (10.3)
Staining—*n* (%)	6 (8.8)
Inter-observer agreement	κ = 0.821 (95% CI, 0.767–0.921)

Abbreviations: CDI, color Doppler imaging; PDI, power Doppler imaging; MI, microvascular imaging; IQR, interquartile range. Note: Pattern percentages are calculated only within MI-positive lesions (*n* = 68); *p*-value from Mann–Whitney U test. Lesion characteristics and microvascular imaging (MI) outcomes.

## Data Availability

The data supporting the findings of this study are available from the corresponding author (H.B.) upon request. The data are not publicly available due to privacy and legal reasons.

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
