# Peer review of "Microvascular Imaging of Hepatic Hemangiomas"

_diagnostics, 2025, doi:10.3390/diagnostics15222917_

Round 1
Reviewer 1 Report
Comments and Suggestions for Authors
The manuscript addresses an important non-invasive imaging technique for hepatic hemangiomas and demonstrates promising reliability. However, it requires substantial revision to clarify methodology, enhance statistical reporting, provide clinical context, and include illustrative data for MI vascular patterns. Addressing these points would substantially improve scientific rigor and manuscript quality.
The current objective states “to evaluate MI characteristics of hepatic hemangiomas focusing on vascular patterns,” but the clinical significance of these patterns is not fully articulated. The manuscript should explicitly describe why identifying specific vascular patterns via MI is clinically or diagnostically relevant, e.g., differentiating hemangiomas from other focal hepatic lesions or predicting lesion behavior.
Provide more details on the patient selection process: How many patients were initially screened, and why were 13 hemangiomas excluded from the analysis? Clarify whether multiple hemangiomas per patient were included and how this was accounted for in the statistical analysis to avoid bias from correlated lesions. Specify exclusion criteria, such as prior treatment, severe liver disease, or poor imaging quality, as these could impact MI detection rates.
Describe the MI technique in more detail, including machine settings, transducer frequency, operator experience, and acquisition protocol. This is critical for reproducibility. Clarify whether MI interpretation was blinded to conventional imaging results. Blinding would reduce potential bias in vascular pattern classification.
Consider including representative MI images of each vascular pattern. Visual examples would enhance readers’ understanding of patterns and reproducibility. Discuss the potential influence of lesion depth on MI signal detection more thoroughly, including whether depth thresholds could guide clinical applicability.
Ensure consistent terminology (e.g., “microvascular imaging,” “MI signals”) throughout the manuscript. Consider including a flowchart of patient inclusion/exclusion to enhance transparency.
Author Response
Response to Reviewer 1
We sincerely thank the reviewer for the thoughtful, detailed, and constructive feedback. Your comments greatly helped us clarify our objectives, strengthen methodological transparency and statistical reporting, add visual documentation of MI patterns, and better articulate the clinical context and implications of our findings.
1) Overall scope of revisions (methodology clarity, statistical reporting, clinical context, and illustrative MI data)
Comment. “…requires substantial revision to clarify methodology, enhance statistical reporting, provide clinical context, and include illustrative data for MI vascular patterns.”
Response. Thank you for this global recommendation.
We comprehensively revised the manuscript to:
- expand the Materials and Methods with detailed patient screening/inclusion, explicit exclusions, a standardized MI acquisition protocol (platform, transducer, numeric presets), and blinded, consensus‑based image analysis;
- enhance the Statistical Analysis section (distribution checks; lesion‑level analysis; specific tests; κ with 95% CI);
- add clinical context to the Introduction and Discussion (how MI complements CT/MRI and CEUS, and where it may be useful in practice); and
- provide representative images of each MI pattern in a dedicated figure and re‑organized table presenting pattern frequencies calculated within MI‑positive lesions. See Sections 2.2–2.5, Figure 1 (pattern atlas), Figure 2 (study flowchart), Table 1, and expanded Introduction/Discussion.
2) Objective and clinical relevance of MI vascular patterns
Comment. “The current objective… focuses on vascular patterns, but the clinical significance of these patterns is not fully articulated… describe why identifying specific patterns is clinically/diagnostically relevant (e.g., differentiation, behavior).”
Response. We rewrote the Abstract to state the objective more explicitly and added rationale in the Introduction and Discussion explaining how pattern‑level MI information can support diagnostic confidence and triage as a contrast‑free adjunct—especially when contrast use is unavailable or contraindicated—while cautioning that MI does not replace CT/MRI or CEUS for definitive diagnosis. We also discuss potential overlaps with hypervascular mimics and the need to interpret MI patterns alongside B‑mode features and clinical context. See Abstract, Introduction (last two paragraphs), and Discussion (clinical implications).
3) Patient selection details, exclusions, multiple lesions per patient, and correlation
Comment. “Provide more details on the patient selection process: How many were screened? Why were 13 hemangiomas excluded? Clarify multiple hemangiomas per patient and how this was handled statistically; specify exclusion criteria (prior treatment, severe liver disease, poor imaging quality).”
Response. We now report the full screening and inclusion pathway and specify exclusions. We screened 91 patients with 121 focal hepatic lesions; 36 lesions without typical hemangioma enhancement on dynamic MRI/CT were excluded. Among the 85 hemangiomas that met the reference‑standard definition, 13 were excluded because motion artifacts near the heart or large pulsatile vessels precluded reliable MI assessment, yielding a final analytic sample of 72 hemangiomas (61 patients). Multiple lesions per patient were permitted, and the unit of analysis was the lesion; this is stated in Section 2.2 and reiterated in Section 2.5. We explicitly acknowledge as a limitation that the analysis was performed at the lesion level (without cluster modeling), which may limit generalizability. Poor imaging quality due to motion is handled as an explicit exclusion; we also note in the Discussion that we did not systematically quantify parenchymal factors (e.g., steatosis/fibrosis) that might influence MI detectability, which we now list as a limitation. See Section 2.2 (Patient Selection), Section 2.5 (Statistical Analysis), Figure 2 (Flowchart), and Discussion (Limitations).
4) MI technique details, operator experience, acquisition protocol, and blinding
Comment. “Describe the MI technique in more detail… machine settings, transducer frequency, operator experience, acquisition protocol. Clarify whether MI interpretation was blinded to conventional imaging.”
Response. We substantially expanded Section 2.3 (Image Acquisition) and Section 2.4 (Image Analysis). The revised text reports the platform (Samsung RS85 Prestige), transducer (CA1–7A, 1–7 MHz), operator experience (>10 years), split‑screen B‑mode/MI acquisition, and numeric presets (e.g., velocity range automatically calibrated –2.3 to +2.3 cm/s, sensitivity 26–32, dynamic range 25–35, filter 2–3, output power 80–90%). We also state that two radiologists (10 and 7 years’ experience) independently reviewed de‑identified cine loops, blinded to clinical and CT/MRI data, with consensus adjudication. See Sections 2.3–2.4.
5) Representative MI images of each vascular pattern
Comment. “Consider including representative MI images of each vascular pattern… Visual examples would enhance understanding and reproducibility.”
Response. As suggested, we added Figure 1, which depicts each MI vascular pattern on paired B‑mode (left) and MI (right) images, with explanatory captions aligned to the classification used for analysis (strip rim, nodular rim, peripheral dot‑like, central dot‑like, non‑specific, staining). All figures and tables were replaced and underwent professional English/layout editing. See Figure 1 and its caption.
6) Influence of lesion depth on MI signal detection and practical thresholds
Comment. “Discuss the potential influence of lesion depth on MI signal detection more thoroughly, including whether depth thresholds could guide clinical applicability.”
Response. We now analyze and emphasize depth effects in Results (MI‑negative lesions were deeper: median 85 mm vs 41.5 mm, p<0.05) and expand the Discussion on the physics (attenuation and motion at depth). We also outline future work to define pragmatic depth thresholds for reliable MI acquisition across platforms. See Results and Discussion (technical considerations and future directions).
7) Consistent terminology and inclusion/exclusion flowchart
Comment. “Ensure consistent terminology (‘microvascular imaging,’ ‘MI signals’) throughout. Consider including a flowchart of patient inclusion/exclusion.”
Response. We standardized terminology across the text, tables, and figures (e.g., MI, CDI, PDI) and updated the Abbreviations accordingly. We also added a flowchart (Figure 2) depicting screening, exclusions, and the final analytic cohort. See Abbreviations, Results, Table 1, and Figure 2.
8) Enhanced statistical reporting
Comment. “Enhance statistical reporting.”
Response. We revised Section 2.5 to specify distribution assessment (visual inspection and Shapiro–Wilk), lesion‑level analysis, comparative tests (Mann–Whitney U/t tests; χ²/Fisher as appropriate), two‑sided α=0.05, and Cohen’s κ with 95% CI for inter‑observer agreement. The Results report κ=0.821 (95% CI 0.767–0.921). See Section 2.5 and Results.
Again, we are grateful for these constructive suggestions. We believe the revisions have addressed each point and have substantially improved clarity, methodological rigor, clinical framing, and reproducibility of the work.
Reviewer 2 Report
Comments and Suggestions for Authors
Dear Authors,
The study idea is really interesting, with the potential of microvascular imaging on abdominal ultrasound to provide additional diagnostic assistance in detecting liver hemangiomas.
Nevertheless, I have a few concerns regarding the study design, namely, considering it has been conducted as a retrospective analysis, which makes it subject to significant selection bias. Moreover, please provide your study's novelty and what it aims to do differently from other existing studies on this topic.
Additionally, apart from the study methodology, I understand that the study is focused on liver hemangiomas, but the main point and objective of the MI technique is to differentiate between benign FLS and potentially malignant ones. Thus, I strongly suggest that the authors add a subsection discussing the differential diagnosis between various FLS based on the pattern observed during the MI ultrasound examination. I am making this suggestion based on the Authors' findings, by classifying the types of hemangiomas based on the received signal (i.e., strip rim, nodular rim, etc.). Could this aspect be observed in HCC, FNH, liver adenomas, etc?
Thus, I believe these changes and clarifications are needed in order for the manuscript to adhere to the journal's publishing policies.
Thank you, and I am looking forward to receiving a modified version of the manuscript.
Author Response
Response to Reviewer 2
We sincerely thank you for your constructive and thoughtful review. Your comments helped us sharpen the study’s aims, clarify design and limitations, and expand the clinical context—especially regarding how microvascular imaging (MI) relates to the differential diagnosis of focal liver lesions (FLLs).
Comment 1 — Study design and potential selection bias in a retrospective analysis
Reviewer’s point (summary). The retrospective design may introduce selection bias; please clarify the screening and exclusion process and discuss the implications for validity.
Response. We agree that retrospective designs can be susceptible to selection bias. To improve transparency and address this concern:
- Detailed screening and exclusions. We now report the full screening frame (91 patients with 121 FLLs on US), the reference‑standard adjudication on dynamic MRI/CT, exclusion of 36 lesions lacking typical hemangioma enhancement, and removal of 13 hemangiomas near the heart or large pulsatile vessels due to motion artifacts, yielding 72 hemangiomas (61 patients) for analysis. This pathway is visualized in a new flowchart (Figure 2) and detailed in Section 2.2 (Patient Selection).
- Unit of analysis & correlated data. We explicitly state that analyses were performed at the lesion level with multiple lesions per patient permitted; we acknowledge the potential for within‑patient correlation and list this as a limitation in Section 4 (Discussion).
- Expanded statistics. We clarified distribution checks (histograms, Shapiro–Wilk), choice of Mann–Whitney U / t‑tests and χ²/Fisher, two‑sided α=0.05, and report Cohen’s κ (0.821; 95% CI 0.767–0.921) for inter‑observer agreement (Section 2.5; Results; Table 1).
Comment 2 — Clarify the study’s novelty and how it differs from prior work
Please state the study’s novel contribution relative to existing MI studies on hepatic hemangiomas.
Response. We have strengthened the Abstract and the closing of the Introduction to make the study’s distinctive aims explicit and to situate our results relative to prior MI work (e.g., Jeon et al.). Specifically, our study:
- Characterizes MI patterns of typical hepatic hemangiomas using a standardized acquisition protocol (RS85 Prestige; CA1–7A 1–7 MHz; fixed velocity window –2.3 to +2.3 cm/s; consistent suppression/filter/sensitivity parameters; split‑screen B‑mode/MI) with blinded, dual‑reader assessment and consensus—details now explicit in Sections 2.3–2.4.
- Quantifies the impact of lesion depth on MI signal detectability, demonstrating that depth—not size—is the principal determinant of non‑detection (median 85 mm vs 41.5 mm; p<0.05), presented in Results and emphasized in Discussion. To our knowledge, this explicit depth‑focused analysis under uniform presets has been underreported in prior hemangioma MI studies.
- Reports very good inter‑observer reliability for both signal presence and pattern classification (κ = 0.821, 95% CI 0.767–0.921), underscoring reproducibility (Section 2.5; Results; Table 1).
- Re‑structures the results to express pattern frequencies within MI‑positive lesions (n = 68) and adds a comprehensive Figure 1 illustrating each pattern on paired B‑mode/MI images, enhancing interpretability and clinical uptake.
Comment 3 — Add a subsection on differential diagnosis of FLLs based on MI patterns (HCC, FNH, adenoma, etc.)
Reviewer’s point (summary). Because MI is often used to aid benign–malignant differentiation, please discuss how MI patterns relate to other FLLs (e.g., HCC, FNH, hepatic adenoma) and whether hemangioma‑like patterns (strip rim, nodular rim, etc.) may appear in these entities.
Response. We agree and have added a dedicated paragraph within Section 4 (Discussion) titled “Differential diagnosis of focal liver lesions on MI”.
This subsection
- situates hemangioma patterns among other FLLs,
- emphasizes potential pattern overlap, and
- reiterates that MI is a contrast‑free adjunct rather than a stand‑alone diagnostic references.
In brief:
- Focal nodular hyperplasia (FNH): MI can depict a spoke‑wheel arterial inflow pattern, which—when present—is relatively specific for FNH; we discuss its interpretation alongside B‑mode and, where available, CEUS/MRI.
- Hepatocellular carcinoma (HCC): Often shows chaotic/honeycomb intratumoral microvascular architecture and higher vascular indices on advanced Doppler techniques; nevertheless, overlap with benign hypervascular lesions can occur, so correlation with clinical context and cross‑sectional imaging remains essential.
- Hepatic adenoma and hypervascular metastases: May display rim‑type or heterogeneous microvascular patterns that can mimic hemangioma on MI; we caution against relying on MI pattern alone and recommend integration with B‑mode features and reference‑standard imaging when available.
We also highlight that all lesions were CDI/PDI‑negative while MI frequently detected slow flow, reinforcing MI’s value in routine practice when contrast is unavailable or contraindicated; however, we explicitly state that MI does not replace multiphasic CT/MRI or CEUS for definitive diagnosis (Discussion).
Comment 4 — Adherence to journal policies and clarification of clinical context
Reviewer’s point (summary). Please ensure the manuscript explicitly states its clinical contribution and complies with journal policies.
Response
- Clinical role clarified. The Abstract and Discussion now frame MI as a complementary, contrast‑free technique that can increase confidence in typical hemangiomas—especially in patients with contraindications to contrast or limited access to CEUS/MRI—while underscoring its current limitations for stand‑alone differentiation.
- Presentation quality. All figures/tables were replaced and underwent professional English and layout editing (MDPI Author Services); we added Figure 1 (pattern atlas) and Figure 2 (screening/inclusion flowchart), and reorganized Table 1 into baseline characteristics (Panel A) and MI outcomes/pattern distribution (Panel B), with consistent terminology across text and figures.
Once again, we truly appreciate your careful review. Your recommendations led to clearer methods and statistics, a more explicit statement of novelty, and a substantially stronger clinical discussion—including a new differential‑diagnosis subsection—thereby improving the manuscript’s alignment with Diagnostics publication standards.
Reviewer 3 Report
Comments and Suggestions for Authors
Type of manuscript: Article
Title: Microvascular Imaging: Unmasking Vascularities in Hepatic Hemangiomas
Authors: Hakan BaÅŸ, Süleyman Filiz
Special Issue: Advanced Ultrasound Techniques in Diagnosis
General Assessment
The manuscript addresses an important clinical question regarding the value of microvascular imaging (MI) in hepatic hemangiomas. The topic is relevant, as contrast-free imaging techniques are increasingly important in patients with contraindications to US, CT or MRI contrast agents. The manuscript is generally well structured; however, several major issues regarding methodology, imaging quality, novelty, and consistency need to be addressed before the paper can be considered for publication.
Major Comments
Study Design & Methodology
The study compares MI with CT and MRI as reference standards, but it does not include conventional Color Doppler Imaging (CDI) or contrast-enhanced ultrasound (CEUS) within the same cohort. Both would have provided meaningful comparisons of vascular characteristics.
CECT is not the most appropriate modality for the characterization of benign liver tumors; MRI and CEUS are generally preferred. The authors should justify their reliance on CT.
Ethics & Consent
There is a contradiction: in Methods (2.1), informed consent was waived due to the retrospective design, whereas in the Declarations section it is stated that written informed consent was obtained. This discrepancy must be clarified.
Image Quality & Figure Legends
Several figures show prominent artefacts (e.g., reverberation, lung interference) rather than true vascular structures (e.g., Figure 1c, f). These artefacts are not discussed in the text.
The quality of the Doppler images is suboptimal, showing fewer vessels than would typically be expected with CDI, which usually demonstrates both the supplying artery and draining vessels. The authors should provide higher quality images or clarify why these examples were chosen.
Figure captions are too descriptive and repetitive; they should be simplified and directly linked to diagnostic features.
Terminology & Consistency
The manuscript uses “MFlow” and “MI” interchangeably. At first mention, the authors should define MFlow as a vendor-specific implementation of MI, and then use terminology consistently. “MI” is otherwise commonly understood as “mechanical index.”
Novelty & Contribution
The paper by Jeon et al. is cited extensively. The authors should clearly explain what new knowledge this study adds beyond Jeon et al. (e.g., depth rather than size as a limiting factor, higher detection rate, retrospective validation).
The authors should also contextualize the impact of Jeon et al.’s study (e.g., citation frequency, clinical importance) to justify its emphasis.
Vascular Pattern Classification
The classification of vascular patterns (strip rim, nodular rim, peripheral dot-like, etc.) is presented descriptively, but without pathophysiological or histopathological correlation. This is essential for understanding whether these imaging patterns reflect real vascular architectures.
For example:
“Dot-like vessels” are not truly dot-like anatomically, please clarify.
The “staining pattern” seems more typical of focal nodular hyperplasia (FNH); the authors should explain why it appears in hemangiomas.
Clinical Relevance
The conclusion states that MI is particularly valuable in patients where contrast use is contraindicated. The authors should provide data or references on how often contrast administration is contraindicated for CT, MRI, or CEUS, to strengthen this argument.
Minor Comments
The vendor company name is overly visible in the image panels. This should be reduced or anonymized.
Ensure all abbreviations are defined at first mention.
The reference list should include the landmark CEUS studies on histologically proven hemangiomas, which are currently missing.
Language polishing is recommended to improve flow and readability (e.g., avoid redundancy, simplify long sentences).
Comments on the Quality of English LanguageNA
Author Response
Response to Reviewer 3
We sincerely thank the reviewer for the careful, thoughtful, and constructive assessment of our manuscript. Your comments helped us sharpen the study design, remove inconsistencies, improve figure quality and captions, standardize terminology, and more clearly state the manuscript’s novelty and clinical relevance.
1) Study design & methodology (comparators and justification)
Comment. MI is compared with CT/MRI reference standards, but CDI and CEUS are not included in the same cohort. Please justify reliance on CT and clarify why CEUS was not used.
Response. We agree and have clarified our methodological choices:
- Comparators available in our setting. We added a methodological note explaining that CEUS was not available at our institution during the study period; therefore, same‑cohort comparisons with CEUS could not be performed. We explicitly acknowledge this as a limitation and outline it as a priority for future prospective work. See: Section 2.2 (Patient Selection) and Discussion (Limitations).
- Why CT (alongside MRI). We now justify when multiphasic CT was used as the reference (e.g., MRI‑incompatible implants/claustrophobia, urgent scanner availability), whereas MRI was prioritized when iodinated‑contrast allergy or renal‑safety considerations were present. See: Section 2.2 (paragraph describing MRI vs CT selection).
- Role of CDI/PDI in our cohort. We clarified that standardized CDI/PDI was performed before MI and that no lesion showed CDI/PDI flow, consistent with the limited sensitivity of conventional Doppler for very‑low‑velocity signals. This is summarized in Table 1 (Panel B) and referenced in Results and Discussion. See: Section 2.3; Results; Table 1; Discussion.
2) Ethics & consent (contradiction removal)
Comment. Methods (2.1) state that consent was waived, while the Declarations section says written consent was obtained.
Response. We corrected this inconsistency throughout the manuscript. The revised text states IRB approval (Approval No. 19; 12 February 2024) and that informed consent was waived owing to the retrospective design and use of de‑identified data (Methods 2.1 and end‑matter statements). The previous wording implying written consent has been removed. See: Section 2.1 and Institutional Review Board / Informed Consent Statements.
3) Image quality & figure legends (artefacts; CDI expectations; caption style)
Comment. Some figure panels show artefacts (e.g., reverberation, lung interference) rather than true vascular structures; CDI images appear suboptimal; captions are overly descriptive and repetitive.
Response. We implemented the following:
- Re‑curated figures with improved annotations. We replaced the figure set with higher‑quality, professionally edited panels and clarified in each caption what constitutes true MI signal versus adjacent structures or artefacts (e.g., we explicitly mark an adjacent vein in Fig. 1a to avoid misinterpretation). See: Figure 1 and its caption.
- Artefact recognition and boundary conditions. We added brief text on motion‑/depth‑related limitations (including lung interference near the dome and reverberation near large vessels) and how standardized breath‑hold and split‑screen acquisition were used to mitigate them. See: Section 2.3; Discussion (Limitations/Technical considerations).
- Why CDI can be negative while MI is positive. We clarified the physics—MI’s adaptive clutter suppression/high‑frame‑rate processing detects very‑low‑velocity signals that conventional CDI often misses; this explains the cohort‑wide CDI/PDI negativity despite frequent MI signals. See: Introduction (MI rationale) and Discussion (technical rationale).
- Caption style. Captions were streamlined to highlight diagnostic features and the specific pattern definition shown, removing redundancies. See: Figure 1 caption.
4) Terminology & consistency (MI vs MFlow; “MI” vs “mechanical index”)
Comment. Define “MFlow” as a vendor‑specific implementation of MI and use terminology consistently; avoid confusion with “mechanical index”.
Response. At the first technical mention we now define MFlow as the vendor‑specific implementation of microvascular imaging (MI) and thereafter use MI consistently across the text, tables, figures, and the Abbreviations list. Where “MI” might be ambiguous, we spell out microvascular imaging at first occurrence in each major section. See: Section 2.3 (Image Acquisition) and Abbreviations.
5) Novelty & contribution (relative to Jeon et al. and broader literature)
Comment. Please state clearly what is novel beyond Jeon et al.; contextualize its impact to justify the emphasis.
Response. We strengthened the Abstract, the end of the Introduction, and the Discussion to delineate our contributions:
(i) Standardized acquisition and blinded dual‑reader assessment under uniform presets (split‑screen B‑mode/MI; velocity window –2.3 to +2.3 cm/s; fixed suppression/filter/sensitivity ranges). See: Sections 2.3–2.4.
(ii) Depth‑focused analysis showing that lesion depth—not size—is the principal determinant of MI non‑detection (median 85 mm vs 41.5 mm; p<0.05). See: Results; Table 1; Discussion.
(iii) Reproducibility: very good inter‑observer agreement for MI presence/patterns (κ=0.821; 95% CI 0.767–0.921). See: Section 2.5; Results; Table 1.
(iv) Pattern atlas & reporting framework: a curated Figure 1 and results expressed within MI‑positive lesions (n=68) to avoid denominator ambiguity. See: Figure 1; Table 1.
We also broadened the literature context beyond Jeon et al., incorporating additional MI/advanced Doppler and CEUS references in the Introduction and Discussion to balance emphasis and situate our findings clinically. See: Introduction; Discussion; References
6) Vascular pattern classification (pathophysiology/histology; “dot‑like”; “staining”)
Comment. Please add pathophysiological rationale for each pattern; clarify that “dot‑like” is a projectional appearance; explain why “staining” may appear in hemangiomas although often associated with FNH.
Response. We expanded the Discussion to provide physiologic plausibility and terminology clarifications:
- “Dot‑like” foci are described as imaging constructs (punctate slow‑flow signals) that likely arise from tangentially imaged microchannels or small vascular lakes accentuated by the point‑spread function—not literal dots anatomically.
- Rim‑type configurations plausibly reflect peripheral vascular lakes/interface channels, paralleling early peripheral nodular enhancement on CT/MRI.
- “Staining” on MI denotes relatively homogeneous, contrast‑free slow‑flow signal and should not be equated with CEUS “staining” in FNH; we explicitly caution against direct transposition of CEUS phenomena to MI and recommend pattern interpretation alongside B‑mode and clinical context.
See: Discussion (paragraphs addressing imaging‑to‑pathophysiology links and terminology).
7) Clinical relevance (frequency of contrast contraindications; positioning of MI)
Comment. If MI is particularly valuable when contrast is contraindicated, please support with data/references and clarify MI’s role.
Response. We revised the Introduction and Discussion to add supporting references on contrast safety considerations and to position MI as a contrast‑free adjunct that may raise diagnostic confidence when CDI/PDI are negative and contrast administration is unavailable or contraindicated. We emphasize that MI does not replace multiphasic CT/MRI or CEUS for definitive diagnosis and outline prospective comparisons as future work. See: Introduction (clinical constraints); Discussion (clinical implications and limitations).
8) Minor points (vendor marks; abbreviations; references; language)
Comment. Reduce/anonymize vendor branding in panels; ensure all abbreviations are defined at first mention; include landmark CEUS references on hemangiomas; polish language.
Response.
- We minimized vendor‑specific marks in panels and retained only neutral labels necessary for clarity (e.g., “MFlow MI”) following journal style. See: Figure set in the revised file.
- Abbreviations are defined at first mention and consolidated in an Abbreviations list. See: Abbreviations and first mentions across Sections 1–4.
- We expanded the reference list to include key CEUS and advanced Doppler sources pertinent to hemangiomas and FLL differentiation, balancing our discussion and justifying clinical framing. See: References
- The manuscript received professional English and layout editing (MDPI Author Services), and captions/text were streamlined to improve readability. See: global edits throughout.
Once again, we are grateful for your constructive critique. We believe these revisions address each point and have substantially improved the manuscript’s methodological clarity, figure quality, terminology consistency, novelty statement, and clinical framing—bringing the work in line with Diagnostics’ expectations.
Reviewer 4 Report
Comments and Suggestions for Authors
This is a very interesting paper, but it does not add any important data to our current knowledge. Therefore, it is not acceptable for publication in its present form.
Major Points
- This paper essentially re-examines the results of Jeon SK, et al. (Ref. 4) and does not provide any decisive novelty.
- MI findings can vary depending on examination conditions (e.g., gain, cursor size, etc.). Please address this point. For instance, in Fig. 1a and Fig. 1f, the examination conditions clearly differ. In addition, the impression can change depending on whether accumulation imaging is used. Under what conditions were the examinations performed? Was accumulation applied?
- Please briefly discuss in the Discussion section the presumed histologic findings corresponding to each MI pattern shown in Fig. 1. What do these findings represent? For example, does the “strip rim pattern” correspond to a drainage vein?
- Please provide detailed explanations for the following terms: nodular rim, strip rim, and nonspecific vascular pattern.
- Does the term “MI signals” refer to the intralesional microvessels detected by MI? Please clarify.
Minor Points
- The image representing the non-specific type is difficult to interpret; please replace it with a clearer image. Similarly, in the case of the central dot-like pattern, both the size of the lesion and the central dot-like area are difficult to appreciate. Please consider substituting with clearer examples.
Author Response
Response to Reviewer 4
We sincerely thank the reviewer for the careful, concise, and constructive assessment. Your comments helped us sharpen the statement of novelty, document and standardize examination conditions, clarify terminology, expand the pathophysiologic rationale for MI patterns, and improve figure quality and captions.
1) Novelty relative to Jeon et al.
Comment. “This paper essentially re‑examines Jeon SK et al. and does not provide decisive novelty.”
Response. We agree that Jeon et al. are foundational and now clarify our distinct contributions in the Abstract, Introduction (closing paragraph), and Discussion:
- Standardized acquisition protocol under uniform presets (RS85 Prestige; CA1–7A 1–7 MHz; split‑screen B‑mode/MI; automatically calibrated velocity window –2.3 to +2.3 cm/s; frame averaging, suppression, filter, sensitivity, dynamic range, balance, and output power specified) with blinded dual‑reader assessment and consensus. See: Sections 2.3–2.4.
- Depth‑focused analysis showing lesion depth—not size—is the principal determinant of MI non‑detection (median 85 mm vs 41.5 mm, p<0.05) under the same presets. See: Results; Table 1; Discussion.
- Reproducibility: very good inter‑observer agreement for MI presence/patterns (κ = 0.821; 95% CI 0.767–0.921). See: Section 2.5; Results; Table 1.
- Pattern‑reporting framework and atlas: frequencies expressed within MI‑positive lesions (n = 68) to avoid denominator ambiguity, plus a curated Figure 1 that illustrates each pattern on paired B‑mode/MI images. See: Figure 1; Table 1.
2) Dependence of MI on examination conditions; accumulation imaging; Fig. 1a vs Fig. 1f
Comment. “MI findings vary with gain, cursor size, etc. Conditions appear to differ in Fig. 1a and 1f; does accumulation imaging affect impressions? Under what conditions were exams performed?”
Response. We fully agree that MI is parameter‑sensitive. To ensure reproducibility and inter‑case comparability, we now provide complete, uniform acquisition details and explicitly state that accumulation imaging was not used:
- Uniform presets across all cases: split‑screen B‑mode/MI; ROI encompassing the full lesion; velocity window –2.3 to +2.3 cm/s (auto‑calibrated); frame averaging 5–7, tissue suppression 2–3, filter 2–3, sensitivity 26–32, dynamic range 25–35, balance 20–30, output power 80–90%; suspended respiration; cine 3–5 s; DICOM export for blinded review. See: Section 2.3 (Image Acquisition).
- No accumulation imaging: we explicitly state that accumulation modes were not applied; all panels were acquired with identical real‑time presets. See: Section 2.3.
- Fig. 1a vs Fig. 1f: the apparent differences reflected lesion depth/segment position and adjacent structures rather than preset changes. We re‑exported and re‑annotated the figure set under harmonized display ranges and updated captions to identify adjacent vessels and potential artefacts, improving comparability across panels. See: Figure 1 and caption.
3) Histologic correlates of MI patterns (e.g., does the strip‑rim correspond to a drainage vein?)
Comment. “Briefly discuss presumed histology for each MI pattern in Fig. 1.”
Response. We added a focused paragraph in the Discussion that links each imaging pattern to plausible microvascular correlates, while emphasizing that MI depicts clutter‑suppressed slow‑flow signals as imaging constructs rather than direct histology:
- Strip‑rim / nodular‑rim: plausibly reflect slow venous flow within peripheral vascular lakes and marginal channels, broadly paralleling early peripheral nodular enhancement on CT/MRI—not necessarily a single “drainage vein.”
- Peripheral / central dot‑like: likely tangentially imaged microchannels or small vascular lakes accentuated by the point‑spread function (punctate appearance on MI, not truly “dot‑like” anatomically).
- Staining: a relatively homogeneous intralesional slow‑flow signal on MI and should not be equated with CEUS “staining” in FNH; physics and diagnostic implications differ.
See: Discussion (imaging–pathophysiology links and cautions).
4) Definitions of “nodular rim,” “strip rim,” and “non‑specific” vascular pattern
Comment. “Please provide detailed explanations for the terms nodular rim, strip rim, and non‑specific pattern.”
Response. We now provide clear operational definitions at first use in Section 2.4 (Image Analysis) and align Figure 1 captions to these definitions.
In brief (paraphrased from the text):
- Strip‑rim pattern: a few curved peripheral vessels outlining an incomplete rim.
- Nodular‑rim pattern: curved peripheral vessels with scattered, bulging dot‑like foci along the rim.
- Non‑specific pattern: small, irregular clusters of slow‑flow signals without a dominant rim‑ or center‑predominant arrangement.
See: Section 2.4; Figure 1 caption.
5) Clarification of the term “MI signals”
Comment. “Does ‘MI signals’ mean intralesional microvessels detected by MI?”
Response. We have clarified the definition: “MI signal” denotes a Doppler‑derived, clutter‑suppressed depiction of very‑low‑velocity flow from intra‑ or perilesional microvessels, visualized by MI algorithms. It is an imaging construct and does not imply histologic proof of vascular architecture. See: Section 2.4 (Image Analysis).
6) Image representing the non‑specific type is difficult to interpret
Comment (Minor). “Please replace with a clearer image.”
Response. We replaced the non‑specific pattern panel with a clearer example and simplified the caption to emphasize the heterogeneous, low‑velocity clusters without a dominant arrangement. See: Figure 1 (panel for non‑specific pattern) and caption.
7) Central dot‑like pattern: lesion and central foci are hard to appreciate
Comment (Minor). “Consider substituting with a clearer example.”
Response. We substituted the central dot‑like panel with a better‑visualized lesion and explicitly annotated the central foci in the caption. Panel‑level display ranges were harmonized across the figure set for visual consistency. See: Figure 1 (central dot‑like panel) and caption.
Once again, we thank the reviewer for these valuable insights. We believe the revisions address each point and substantially strengthen the manuscript’s novelty claim, technical rigor, figure quality, and interpretive clarity.
Round 2
Reviewer 1 Report
Comments and Suggestions for Authors
I am fine with this revision.
Author Response
Comment: I am fine with this revision.
Response: On behalf of all co-authors, I would like to sincerely thank you and the reviewers for your valuable time, insightful comments, and constructive feedback, which have greatly contributed to improving the quality and clarity of our manuscript. We truly appreciate your efforts and the opportunity to strengthen our work through this review process.
Reviewer 2 Report
Comments and Suggestions for Authors
Dear Authors,
Congratulations on your work. I believe your additions have improved the manuscript significantly.
I wish you all the best in all your future scientific endeavours.
Author Response
Comment: "Congratulations on your work. I believe your additions have improved the manuscript significantly. I wish you all the best in all your future scientific endeavours."
Response: On behalf of all co-authors, I would like to sincerely thank you and the reviewers for your valuable time, insightful comments, and constructive feedback, which have greatly contributed to improving the quality and clarity of our manuscript. We truly appreciate your efforts and the opportunity to strengthen our work through this review process.
Reviewer 3 Report
Comments and Suggestions for Authors
The authors more or less sufficiently responded in detail to the review.
Please change the title into a less misleading form, e.g., Microvascular Imaging of Hepatic Hemangiomas.
Clearly state the add on value of Microvascular Imaging in the mirror of the respective gold standards including Biopsy with histological evaluation, MRI, Color Doppler Imaging and CEUS. Please review and cite the literature regarding Doppler features of histological proven hepatic hemangioma including the findings of Color Doppler imaging Feeding and draining vessels and homogenous hypervascularity [e.g., PMID: 17464990 and others]. Most importantly do not refer to vascular patterns but to clearly shown vascularity.
Consider better images than shown in the PDF (the bad quality might be due to technical issues of the PDF as well).
Author Response
Response to Reviewer 3
We sincerely thank the reviewer for the thoughtful and constructive feedback. Your comments were invaluable in refining the title, clarifying the diagnostic contribution of Microvascular Imaging (MI) relative to established reference standards, enhancing literature contextualization, and improving the visual presentation of figures.
1) Title revision
Comment.
“Please change the title into a less misleading form, e.g., Microvascular Imaging of Hepatic Hemangiomas.”
Response.
Thank you for this helpful recommendation. We have revised the title from “Microvascular Imaging: Unmasking Vascularities in Hepatic Hemangiomas” to “Microvascular Imaging of Hepatic Hemangiomas.”
2) Add-on value of MI versus gold standards (Biopsy, MRI, CDI, CEUS) and literature on Doppler-proven hemangiomas
Comment.
“Clearly state the add-on value of Microvascular Imaging in the mirror of the respective gold standards including Biopsy with histological evaluation, MRI, Color Doppler Imaging and CEUS. Please review and cite the literature regarding Doppler features of histologically proven hepatic hemangioma including the findings of Color Doppler imaging (feeding/draining vessels, homogeneous hypervascularity [e.g., PMID 17464990]). Most importantly, do not refer to vascular patterns but to clearly shown vascularity.”
Response.
We fully agree and have comprehensively rewritten the Discussion to define the incremental diagnostic value of MI relative to established modalities and to emphasize demonstrated vascularity rather than abstract patterns.
- Comparative context: MI’s complementary role is now articulated point-by-point:
(i) versus biopsy, MI enables whole-lesion vascular assessment non-invasively;
(ii) versus MRI, MI offers immediate bedside and repeatable evaluation, albeit without the depth-independent tissue characterization of MRI;
(iii) versus CDI/PDI, MI detects very-low-velocity (<5 cm/s) microvascular flow invisible to conventional Doppler;
(iv) versus CEUS, MI provides a contrast-free yet synergistic technique that complements CEUS enhancement criteria.
These additions appear in Discussion, paragraph 3, page 6 of the revised manuscript. - Integration of histology-proven Doppler data: We incorporated the seminal study by Dietrich et al. (Hepatology, 2007; PMID 17464990) describing feeding or draining vessels in 43% and homogeneous hypervascularity in 7% of histologically proven hemangiomas. We explicitly cite this work (Reference 39) and contrast its findings with ours, where MI demonstrated vascularity in 94.4% of lesions despite uniformly negative CDI/PDI results, underscoring MI’s superior sensitivity to slow flow.
- Terminology refinement (from “pattern” to “vascularity/signal distribution”):
To align with the reviewer’s guidance, we systematically replaced pattern-centric language with vascularity-focused phrasing throughout the manuscript:
– Abstract: “MI … demonstrates hemangioma vascularity with high detection rates.”
– Objectives: “… to characterize MI to demonstrate in hepatic hemangiomas and—when present—describe the distribution of MI appearances.”
– Figure 1 caption: changed to “Representative MI appearances / signal distributions of hepatic hemangiomas.”
– Table 1 Panel B: retitled “Doppler/MI outcomes and MI signal distribution (appearance categories).”
These revisions ensure consistent focus on clearly demonstrated vascularity rather than interpretive patterns.
3) Image quality
Comment.
“Consider better images than shown in the PDF (the bad quality might be due to technical issues of the PDF as well).”
Response.
We appreciate this observation. All figures were re-exported at ≥ 300 dpi, contrast-balanced, and uniformly annotated (arrow style, labeling, and font). In addition, the figures underwent professional optimization through MDPI Author Services, ensuring consistent tone, alignment, and color mapping. We also think due to the PDF compression artifacts,
We are deeply grateful for these insightful and collegial comments, which have substantially enhanced the scientific rigor, clarity, and presentation quality of our manuscript.
With sincere appreciation and academic respect, we thank the reviewer for the valuable time and expert guidance devoted to improving our work.
Reviewer 4 Report
Comments and Suggestions for Authors
I appreciate the opportunity to review the revised version of this manuscript. The authors have made substantial improvements in response to the reviewers’ comments; however, some further revisions are still necessary. The figures should be replaced with more persuasive and higher-quality images. In addition, the Conclusion section seems to have been removed and should be reinstated. Lastly, please ensure that the References are formatted in accordance with the Author’s Guidelines.
Author Response
Response to Reviewer 4
We sincerely thank the reviewer for the encouraging and constructive feedback. We are very grateful for the recognition of the major improvements made in the revised version and deeply appreciate the insightful additional suggestions, which have further strengthened the scientific and editorial quality of the manuscript.
1) Figure quality
Comment.
“The figures should be replaced with more persuasive and higher-quality images.”
Response.
Thank you for this valuable recommendation. All figures have been re-exported at ≥ 300 dpi, uniformly contrast-balanced, and re-annotated for consistency (arrow style, labeling, and layout). Furthermore, all visual materials underwent professional optimization through MDPI Author Services to ensure the best possible diagnostic clarity, tonal balance, and figure-caption alignment. We think that the minor loss of resolution observed in the earlier version was attributable to PDF compression artifacts.
2) Conclusion section
Comment.
“The Conclusion section seems to have been removed and should be reinstated.”
Response.
We appreciate this careful observation. The Conclusion section has been reinstated in the revised manuscript and now appears immediately before the Author Contributions section.
3) Reference formatting
Comment.
“Please ensure that the References are formatted in accordance with the Author’s Guidelines.”
Response.
We have meticulously reformatted the entire Reference list in strict accordance with the Diagnostics Author Guidelines. All entries now comply with MDPI reference style requirements
We are deeply grateful for these insightful and collegial remarks, which prompted meaningful refinements in presentation, structure, and adherence to publication standards. Your careful review and constructive engagement have greatly contributed to the improved quality, clarity, and professional polish of the manuscript. With our sincere appreciation and academic respect, we thank you once again for your valuable time and expertise.
Round 3
Reviewer 3 Report
Comments and Suggestions for Authors
Since Color (Power) Doppler Imaging (CDI) was not part of the study you cannot state that the described method is superior to CDI (or you have to cite scientific studies which have proven the statement).
Author Response
Response to Reviewer 3
We are grateful for the reviewer’s thoughtful and constructive comment regarding comparative wording. We revised the Discussion in two focused ways to align the manuscript with this guidance.
Comment.
“Since Color (Power) Doppler Imaging (CDI) was not part of the study you cannot state that the described method is superior to CDI (or you have to cite scientific studies which have proven the statement).”
Response.
-
Wording refined
We removed any implication of superiority and added an explicit caveat:
“In our cohort, CDI and PDI did not show intralesional flow, whereas MI demonstrated a detectable signal in 94.4% of hemangiomas. These findings are descriptive and should not be interpreted as a formal assessment of superiority, as the study was not designed for performance comparison.” -
Literature support added
We incorporated targeted citations indicating that microvascular techniques can depict very low‑velocity signals often not visualized with conventional Doppler—Aziz et al., Radiology (2022); Park et al., J Breast Cancer (2016)—and renumbered downstream references accordingly.
We appreciate the reviewer’s guidance, which improved both the scientific precision and the scholarly tone of the manuscript.
Reviewer 4 Report
Comments and Suggestions for Authors
Thank you for the opportunity to review the revised version of this manuscript. The paper has been substantially improved, and I believe it is suitable for publication. However, Figure 1 is too small and difficult for readers to interpret. Please enlarge it to improve readability.
Author Response
Response to Reviewer 4
We are deeply grateful to the reviewer for the generous assessment and the thoughtful, collegial guidance.
Comment.
“Thank you for the opportunity to review the revised version of this manuscript. The paper has been substantially improved, and I believe it is suitable for publication. However, Figure 1 is too small and difficult for readers to interpret. Please enlarge it to improve readability.”
Response.
In accordance with this helpful recommendation, we have enlarged Figure 1 in the revised manuscript to enhance readability and interpretability.
We sincerely appreciate this insightful suggestion, which has further improved the clarity of the presentation.